

# Human disturbances and the daytime activity of sympatric otters along equatorial Amazonian rivers

Darren Norris[1,2,3,*] and Fernanda Michalski[1,2,4,*]

[1] Postgraduate Programme in Tropical Biodiversity, Federal University of Amapá, Macapá, Amapá, Brazil
[2] Ecology and Conservation of Amazonian Vertebrates Research Group, Federal University of Amapá, Macapá, Amapá, Brazil
[3] Environmental Sciences, Federal University of Amapá, Macapá, Amapá, Brazil
[4] Pro-Carnivores Institute, Atibaia, São Paulo, Brazil
[*] These authors contributed equally to this work.

Corresponding author
Darren Norris, darren.norris@unifap.br

## ABSTRACT

**Background.** Previous studies suggest coexistence between sympatric neotropical (*Lontra longicaudis*) and giant otters (*Pteronura brasiliensis*) maybe facilitated by temporal and spatial differences in activity. Yet, to date there has been no systematic evaluation of activity of these species in sympatry. Here we use extensive multi-year field data to compare temporal and spatial patterns in the diurnal activity of sympatric giant and neotropical otters to answer three questions: Do temporal patterns in daytime river use change in relation to seasonal river levels (low, rising, high and declining river levels), do they change due to human disturbances (boats and fishing nets) and do patterns in neotropical otter activity change due to the presence of the larger sized giant otter?

**Methods.** Direct observations of both species were recorded using standardized boat surveys along 218 km of rivers over 53 months during nine years (2011–2013 and 2015–2020). Complementary techniques (Generalized Additive Models, Kernel density estimates and non-parametric tests,) were used to compare diurnal activity patterns along rivers subdivided into 41 river reaches.

**Results.** The presence of giant otters decreased threefold from 67% of the least disturbed reaches (few boats no fishing nets) to 18% of the most disturbed reaches with many boats and fishing nets. In contrast neotropical otter presence nearly doubled from 44% of the least disturbed to 73% of the most disturbed reaches with fewest giant otter detections. Both species were observed across all daytime hours but were observed rarely on the same day. There was no evidence to suggest simultaneous use of the same reach. When species were detected on the same day, they were separated spatially (median distance between species 12.5 km) and temporally (median time difference 3.0 hours). There was little change in activity of either species among seasons. Giant otters were less active in river reaches with fishing nets and boat use, whereas neotropical otter activity did not appear to be strongly affected by these activities.

**Conclusions.** Our findings support evidence that diurnal activity in both otter species is flexible, with daytime activity changing due to human disturbances in the case of giant otters.

## INTRODUCTION

The expansion of human activities and associated disturbances can change mammal activity patterns (*Gaynor Kaitlyn et al., 2018*; *Suraci et al., 2021*). Impacts are variable, with the strength and direction of effects due to hunting (*Di Bitetti et al., 2008*; *Frey et al., 2020*) and habitat fragmentation causing species specific changes (*Frey et al., 2020*; *Norris, Michalski & Peres, 2010*; *Rheingantz et al., 2016*). For example, the daily activity pattern of red brocket deer (*Mazama americana*) shifted from more nocturnal to diurnal in better protected areas, whereas dwarf brocket deer (*Mazama nana*) showed no difference in their daily activity pattern (*Di Bitetti et al., 2008*). It has been hypothesized that human disruption could increase temporal overlap between carnivores, increasing interspecific competition, yet to date such hypotheses have only been examined in terrestrial carnivores (*Sévêque et al., 2020*; *Sévêque et al., 2021*). While temporal separation has been suggested to facilitate species co-occurrence the activity of sympatric semi-aquatic Neotropical otter species remains poorly studied. To date no studies have documented how activity may change as a result of the rapidly expanding human river use across the range of neotropical (*Lontra longicaudis*) and giant otters (*Pteronura brasiliensis*).

Giant and neotropical otters are widespread and ubiquitous components of Amazonian freshwater systems, but human disturbances and changes to these systems increasingly threatens both species (*Duplaix, Evangelista & Rosas, 2015*; *Rheingantz, Santiago-Plata & Trinca, 2017*). The group living giant otter is the largest species of the Lutrinae subfamily, reaching up to approximately 30 kg and 1.8 m in length (*Duplaix, Evangelista & Rosas, 2015*). The smaller bodied (adults typically weigh less than 12 kg) solitary neotropical otter is sympatric with giant otters across the latter's current distributional range (*Duplaix, Evangelista & Rosas, 2015*; *Rheingantz, Santiago-Plata & Trinca, 2017*). Giant otter populations were decimated by hunting in the 19th and 20th centuries and remain Endangered (A2cde+3ce) with declining populations according to the IUCN (*Groenendijk et al., 2022*). Neotropical otter populations are also declining and classified as Near Threatened (A3c) by the IUCN (*Rheingantz et al., 2022*).

Giant otters are known to be highly sensitive to human disturbances due to factors such as their large body size and social structure (*Barocas et al., 2022*; *Cook et al., 2022*; *Duplaix, Evangelista & Rosas, 2015*; *Michalski & Norris, 2021*; *Pimenta et al., 2018*). Although there are occasional reports of populations recovering from parts of the historic range (*Recharte Uscamaita & Bodmer, 2010*) giant otters are still hunted across Amazonia, including in the Brazilian state of Amapá due to real and/or perceived competition with fishers (*Michalski et al., 2012*). Neotropical otters are also threatened by negative interactions with fishers (*Rheingantz et al., 2022*; *Rheingantz, Santiago-Plata & Trinca, 2017*). Similarly, increasing deforestation, land use change and pollution in and around Amazonian waterways threatens

both species (*Duplaix, Evangelista & Rosas, 2015*; *Groenendijk et al., 2022*; *Rheingantz et al., 2022*; *Rheingantz, Santiago-Plata & Trinca, 2017*).

Although competition between sympatric carnivores has been widely reported there is no evidence of direct competition between giant and neotropical otters (*Duplaix, Evangelista & Rosas, 2015*; *Rheingantz, Santiago-Plata & Trinca, 2017*). Temporal and spatial patterns of river use have been documented for both giant otters (*Duplaix, 1980*; *Leuchtenberger et al., 2014*; *Oliveira, Norris & Michalski, 2015*) and neotropical otters (*Rheingantz et al., 2016*; *Rheingantz, Santiago-Plata & Trinca, 2017*). However, temporal and spatial interactions between these two otter species have been largely overlooked and studies involving both species in sympatry were mainly concentrated in scat surveys to assess feeding behavior and food niche overlap (*Moraes et al., 2021*; *Muanis & Oliveira, 2011*; *Silva, Rosas & Zuanon, 2014*).

River based activity (*e.g.*, swimming, feeding) appears to be primarily diurnal in both giant and neotropical otters (*Duplaix, Evangelista & Rosas, 2015*; *Rheingantz, Santiago-Plata & Trinca, 2017*). The activity of both species is flexible and may change seasonally due to differences in prey/habitat availability (*Duplaix, 1980*) or in response to human disturbances (*Barocas et al., 2022*; *Rheingantz et al., 2016*). Although giant and neotropical otters are both widespread across the neotropics, surprisingly few studies simultaneously evaluate both species. One of the earliest documented reports indicated that both species may occur in the same river stretches and even exploit the same pools during the dry season when there are limited aquatic feeding habitats available (*Duplaix, 1980*). Studies suggest that when sympatric the species coexist based on differences in diet (*Moraes et al., 2021*) and activity, with neotropical otters suggested to be more crepuscular and nocturnal when sympatric with giant otter (*Duplaix, 1980*). Others analyzed scat samples and showed differences in diet between species (*Moraes et al., 2021*; *Silva, Rosas & Zuanon, 2014*). A study from Pantanal wetlands showed neotropical otters to be more generalist in their ecological requirements (*Muanis & Oliveira, 2011*), which is to be expected considering the differences in body size and diet of the two species.

Despite the increase in the number of studies on giant and neotropical otters' biology, behavior and ecology, there is no information published on the influence of seasonal changes in otters activity patterns (*Camp, 2021*; *Duplaix, Evangelista & Rosas, 2015*; *Rheingantz, Santiago-Plata & Trinca, 2017*). To date, most of the scientific literature describing the influence of seasonal changes focused on home range differences, with evidence of giant otters expanding their home range to flooded forest areas during the Amazonian rainy season (*Evangelista & Rosas, 2011*) and home ranges increasing from four to 59 times during the wet season in relation to the dry season in the Brazilian Pantanal (*Leuchtenberger et al., 2013*). Curiously, the literature on neotropical otters does not mention seasonal influences in any aspect of their temporal activity or home range, with only 4% of the studies focusing on behavior and demography in a recent literature review (*De Almeida & Ramos Pereira, 2017*).

Considering the increasing threats to both species across Amazonia, increasing knowledge of the activity and interactions of giant and neotropical otters is vital to help inform appropriate protection and management policies. Our objective was to compare

activity of sympatric otter species along Amazonian rivers with different degrees of human river use. This was achieved by focusing on three questions: Do temporal patterns in daytime river use change seasonally, do they change due to human influences (boats and fishing nets), and do patterns in neotropical otter activity change due to the presence of the larger sized giant otter?

## MATERIALS & METHODS

### Study area

This study was conducted in the Araguari river basin, eastern equatorial Amazonia, in the central region of the Brazilian State of Amapá (Fig. 1). The basin includes rivers classified as "clear water" (*Junk et al., 2011*). The Araguari River has an extension of 560 km (*Ziesler & Ardizzone, 1979*) and rises in the Guiana Shield at the base of the Tumucumaque uplands, bifurcates and discharges into both the Amazon River and the Atlantic Ocean. The region's climate is humid tropical ("Am" Tropical monsoon; *Kottek et al., 2006*), with a dry season from September to November (<150 mm of monthly rain) and the rainy season (>300 mm of monthly rain) from February to April.

The forests surrounding the study rivers are part of the eastern Amazon Guianan forests (*Ter Steege et al., 2001*) and consist predominantly of never-flooded closed canopy tropical rainforest vegetation dominated by members of the Fabaceae, Sapotaceae, Lecythidaceae and Lauraceae (*Batista et al., 2015*). Although deforestation has recently increased in Brazil, Amapá is the state with the lowest deforestation rate within the 5 Mkm$^2$ Brazilian Legal Amazon (*INPE, 2021*; data available from: http://terrabrasilis. dpi.inpe.br/app/dashboard/deforestation/biomes/legal_amazon/increments, accessed 22 November 2021). In our study area there has been some localized deforestation extending approximately 50 km upstream from the nearest town (Porto Grande), yet the majority of survey rivers are immediately bordered by continuous forest cover, *i.e.*, < 1% forest loss from 2000–2020 within 10 km of the rivers (*Hansen et al., 2013*; https://glad.earthengine.app/view/global-forest-change#dl=1;old=off;bl=off;lon=-51.59987769825302;lat=1.0675835988661395;zoom=10, accessed 7 July 2022) with a border of canopy trees typically starting 1–4 m from the river's edge and the riverbank rises abruptly along the studied rivers such that forests close to the margin (*e.g.*, within 110–554 m) are never flooded (*Caron et al., 2021*).

### Ethics statement

This study used data from non-invasive field observations and did not involve direct contact or interactions with animals. Fieldwork was conducted under research permit numbers IBAMA/SISBIO 26653, 49632, and 69342 to DN and FM, issued by the Instituto Chico Mendes de Conservação da Biodiversidade (ICMBio).

### Data collection

Field data was collected over 53 months during nine years (2011-2013 and 2015-2020) along 218 km of rivers. Standardized river based boat surveys were used (*Groenendijk et al., 2005*) with direct observations recording the geographic location and timing of otter

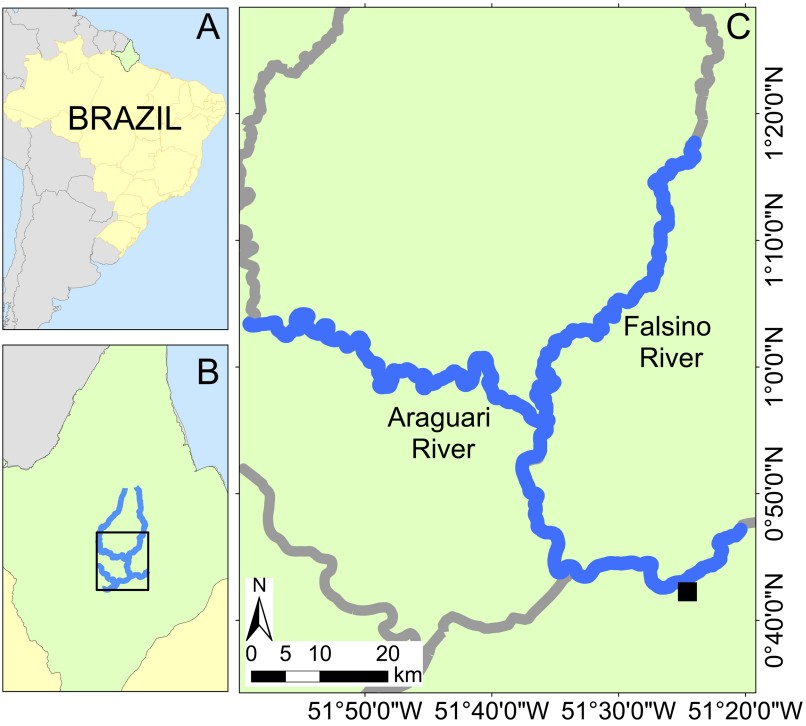

**Figure 1** Study area location with insets showing Brazil (A) and Amapá state (B). Location of the Araguari and Falsino rivers (C) showing surveyed rivers (blue lines) and the nearest town (Porto Grande, black square).

activity in the rivers. The principal objective was to monitor temporal and spatial patterns, not evaluate behavior. Therefore, observations were limited to a 15 min maximum that prevented observation of additional details such as feeding/social behavior that require extended acclimatization of the study species. Diurnal boat surveys at a standard speed (mean ± SD = 9.6 ± 2.8 km/h, min-max = 1.4–15.9 km/h), by a minimum team of two people, were used to search for direct sightings of giant and neotropical otters along the river stretches (*Groenendijk et al., 2005*; *Oliveira, Norris & Michalski, 2015*). To minimize any possible observer related bias between years, a local resident with over 20 years of knowledge on otters, was present in our searches throughout the entire study period. Locations and number of boats and fishing nets during the study period were also obtained during our boat surveys with the aid of a handheld GPS.

## Data analysis

We adopted complementary analytic approaches to understand how season and human disturbances influenced otters diurnal activity *i.e.,* "if" and "when" otters were active along rivers. Firstly, generalized additive models (GAMs) were used to evaluate the relative importance of human influences, season and time of day on the presence of both otter species. Activity patterns were subsequentially compared between species using (i) kernel density estimates (*Ridout & Linkie, 2009*; *Rowcliffe et al., 2014*) and (ii) comparison of relative abundances using non-parametric tests with *P*-values calculated using Bonferroni

corrections for multiple comparisons. A copy of the data used in the analysis is available in Supplementary Material S1.

We did not transform hours as surveys were only conducted during daylight hours and in contrast to known biases at high latitudes (*Vazquez et al., 2019*) there is no daylight saving time and minimal variation in day length at our low latitude study area, where the sun rises at approximately 06:30 am year round (day length is 11.95 h in December and 12.05 h in June at the central latitude of 0.93N). The focus on daytime (diurnal) activity also precluded the need to adopt techniques developed for comparisons across the circular 24 h diel cycle (*Ridout & Linkie, 2009*).

Otter sightings and human river use (boats and fishing nets) were summarized along rivers subdivided into 41 river reaches (continuous sub-segments) of approximately 5 km (mean = 5.3 km, SD = 0.05). Such an approach enabled us to evaluate species activity in relation to variables such as boats and nets that are spatially heterogeneous across the rivers. This approach of dividing river lengths into shorter reaches (sub-sections sensu *Smith et al. (2020)*) has been applied previously for giant otters (*Oliveira, Norris & Michalski, 2015*) and neotropical otters (*Smith et al., 2020*). The decision as to the length of reaches *i.e.,* separation distances between observations is important to avoid issues of spatial autocorrelation. A previous study with giant otters in the same region (*Oliveira, Norris & Michalski, 2015*) obtained an optimal bandwidth (*i.e.,* separation distance; *Berman & Diggle, 1989*; *Hengl, 2009*) of any value greater than 456 m. The final 5 km value was chosen based on a combination of practical and statistical considerations. A length of 5 km corresponds (approximately) to the minimum seasonal giant otter range size of 4.6 km reported along Amazon rivers (*Evangelista & Rosas, 2011*) and is therefore a relevant and informative scale for our understanding of the species responses. Although there are no comparable studies documenting movements or ranging of neotropical otters across Amazonia (*Camp, 2021*), a recent study also suggests a length of 5–10 km for spatially independent registers of this species (*Smith et al., 2020*). Additionally, a short-term (35 day) radio-telemetry study from southeastern Brazil showed a single male neotropical otter moved 1–2.6 km per day around coastal islands (*Nakano-Oliveira et al., 2004*). The number of independent replicate river reaches ($n = 41$) provided sufficient information to evaluate variation in the species activity and human disturbances. This number of river reaches enabled us to capture spatial variability yet was not at such a fine scale as to generate zero inflation problems (*i.e.,* finer scales generate severely "inflated" proportion of reaches with zero otter encounters and zero records of houses/boats *etc.*).

Records from all years were summarized for each hour and river level per 5 km river reach to address each of the three questions. This was necessary as there were insufficient observations to model finer scale variation among years within reaches. To compare activity among seasons, otter sightings were summarized among river levels grouped into four classes: low (September to November), rising (December, January, February), high (March to May) and decreasing (June to August). Associations with otter activity and human influences were compared by grouping river reaches by the relative intensity of human river-based activity. Reaches were classified by boat frequency (number of boats per survey km). Those with none or relatively few boats (less than 0.006 boats per survey km the

25% quantile of density values), intermediate (25–75% quantile range of density values) and many (greater than 0.08 boats per survey km the 75% quantile of boat frequency values). Diurnal activity patterns were also compared between reaches with ($n = 26$) and without ($n = 15$) fishing nets during the nine years. Although it was not logistically possible to obtain equal survey effort across all the different factor levels, our extensive survey efforts provided a representative sample across the 41 reaches (Supplemental Material S2). While there was variation in effort among different reaches (mean ± SD days per reach = 65.9 ± 31.6), overall, there was greater survey effort (km per river reach; Fig. S2.2) in areas with more anthropogenic impacts. Therefore, any reduction in otter activity cannot be explained by differences in effort.

Generalized additive models (GAMs) were used to evaluate the relative importance of human influences, season and time of day between otter species. GAMs are an extension of generalized linear models but have important and fundamental differences in their estimation as they relax parametric assumptions (Wood, 2017). GAMs are a powerful and flexible modeling technique that provides a systematic description of the patterns in the data rather than focusing solely on the statistical significance of the differences between the response and explanatory variables (Pedersen et al., 2019; Van Rij et al., 2019; Wood, 2017). GAMs were chosen to develop models explaining patterns in the data as the responses of otter encounters (presence/absence) could be modelled against covariates using a combination of parametric, non-parametric (smoothed non-linear) and random terms (Pedersen et al., 2019; Wood, 2017). As such GAMs also enabled us to control spatial and temporal variation and autocorrelation and provided a more robust and comprehensive evaluation of spatial and temporal patterns than the subsequent kernel density and non-parametric tests in isolation. The modelling approach adopted followed guidance and recommendations presented by Pedersen et al. (2019), Van Rij et al. (2019) and Wood (2017).

GAMs modelled how the probability of encountering the otter species varied with season (river level), time of day (hour), boat frequency, and the presence of fishing nets (Table 1). Summaries for each hour and river level per 5 km river reach provided a total of 1296 presence-absence observations for the GAMs. To control for differences in survey effort GAMs included total km as an observation level offset term. Species were modelled separately, and the presence of giant otters was also included in the model of neotropical otter encounters. Binomial response (presence/absence) was modelled with the quasibinomial distribution family to account for overdispersion (i.e., a lack of independence so that extra-binomial variation occurs) and zero-inflation in the data; with models estimated using restricted maximum likelihood (REML; Anderson, Burnham & White, 1994; Pedersen et al., 2019). Further details of GAM model specifications are presented in Supplemental Material S3.

Information theoretic model selection (Burnham & Anderson, 2002) was used to establish the relative importance of the different variables. Models are ranked and scaled by an information criterion to allow an understanding of model uncertainty over the set of candidate models (Burnham & Anderson, 2002: pp. 281–284). We evaluated models based on their information content, as measured by QAIC Quasi-Akaike Information Criterion

**Table 1  Variables used to explain variation in otter activity.** Variables included in Generalized Additive Models to explain activity (presence-absence) of giant and neotropical otters along 41 river reaches.

| Variable | Value type | Variable description |
|---|---|---|
| Boat frequency | Categorical factor with three levels. | Reaches were classified by boat frequency (number of boats per survey km). Those with none or relatively few boats (less than 0.006 boats per survey km the 25% quantile of frequency values), intermediate (25–75% quantile range of frequency values) and many (greater than 0.08 boats per survey km the 75% quantile of boat frequency values). |
| Fishing net presence | Categorical factor with two levels. | Presence or absence of fishing nets in the river reach. |
| Giant otter presence | Categorical factor with two levels. | Presence or absence of giant otter in the river reach. Only used to model neotropical otter activity. |
| Location | Continuous | Geographic coordinates of the center of each river reach. |
| Season—river level | Continuous | River levels grouped into four discrete values: "1": low (September to November), "2": rising (December, January, February), "3": high (March to May) and "4": decreasing (June to August). |
| Time—hour of day | Continuous | Includes all survey hours with and without detections. |
| Effort—survey km | Continuous | Survey effort corresponding to each observation. |

implemented in R package 'MuMIn' (*Barton, 2022*; *Bolker, 2022*). QAIC was adopted as global models for both species were overdispersed, *i.e.,* the global model residual deviance was greater than residual degrees of freedom (*Burnham & Anderson, 2002*: pp. 66–70; *Crawley, 2007*: pp. 528).

A subset of models excluding highly correlated variables was used for information theoretic model selection. Models with both fishing nets and boats were excluded for giant otters. Models with a combination of fishing nets, boats and/or giant otter were excluded in the case of neotropical otters. With a strong a priori justification for inclusion, we retained all subset models and all variables; therefore, all predictors were on equal footing to calculate their relative importance as measured by the variables' Akaike weights (*Burnham & Anderson, 2002*: pp. 75–77, 167–172), which is a scaled measure of the likelihood ratio that ranges between 0 (least important) and 1 (most important). This approach also enabled us to compare different model specifications including hierarchical specifications representing different assumptions about the degree of intergroup variability in otter species functional response (*Pedersen et al., 2019*). Variable importance was obtained from a model subset including models having a ΔQAIC-value ≤ 6 (*Richards, 2008*). Although ΔAIC values ≤2 are often adopted to select models, simulations show ΔAIC values ≤6 are more reliable when the true effect of a measured factor is relatively weak, data are highly overdispersed and/or data are few (*Richards, 2008*). These issues also limit the ability to reliably estimate effect size and direction particularly in the case of observational data such as ours. Considering the small sample sizes (75 giant otter and 65 neotropical otter detections) and large proportion of zeros, we limited our GAM analysis to comparing the relative importance of variables between otter species.

Activity patterns were then compared between species using (i) kernel density estimates (*Ridout & Linkie, 2009*; *Rowcliffe et al., 2014*) (ii) the distribution of relative abundances.

Kernel density estimates were obtained using calculations bounded by daytime hours and weighted by the overall effort (km) at each hour for each river reach to account for differences in detectability (*Rowcliffe et al., 2014*). Overlap in activity was tested *via* a randomisation test for the probability that two sets observations come from the same distribution and activity levels compared using a Wald test for the statistical difference between activitiy level estimates (*Ridout & Linkie, 2009*; *Rowcliffe et al., 2014*).

The distribution of relative abundances was compared using sightings classified by time of day grouped in 3-hour intervals (early morning, 06:00-08:59; late morning, 09:00-11:59; early afternoon, 12:00-14:59; and late afternoon, 15:00-17:59). To avoid spatial correlations (Supplemental Material S2) in the comparison of relative abundances boat frequency and net presence were combined into a single "human-use" factor with four levels: few, intermediate no nets, intermediate with nets and many. Finally, to examine if patterns in neotropical otter activity changed due to the presence of giant otters, the activity of neotropical otters was also compared among the 25 reaches with neotropical otters and reaches with ($n = 14$) and without ($n = 11$) giant otter detections. To test for differences in mean values (Kruskal-Wallis rank sum test) and distribution (Two-sample Kolmogorov–Smirnov test) among groups we used *P*-values calculated using Bonferroni corrections.

## RESULTS

A total of 140 detections (75 giant otter and 65 neotropical otter) were obtained from 13,399 km and 1,697 h of boat surveys conducted over 397 days during 53 months from 2011 to 2020 (values do not include time spent moving between different regions or with rain). Anthropogenic variables and survey effort were most important in explaining if and when giant otters were active (Fig. 2). For neotropical otters survey effort, giant otter presence, and the interactions of giant otter presence with river level season and time of day were most important (Fig. 2). Giant otter presence declined with increasing boat frequency. Indeed, boat frequency was the only factor with a level whose model averaged confidence intervals did not overlap zero for giant otters (95% CI many boats: $-3.53$ to $-0.92$). In contrast, for neotropical otters the confidence intervals of all factor levels overlapped zero.

### Seasonal patterns in diurnal activity

Both otter species were detected during all daytime hours and all seasons (Figs. 3 and 4). Neotropical otters tended to be more active earlier in the day compared with giant otters (Fig. 3), but there was only weak statistical difference in activity levels (Wald test $P = 0.010$) and strong overlap in temporal activity (86.2%, $P = 0.214$, Fig. 3). Additionally, there was no significant difference in the overall distribution of daytime activity between species (Fig. 3A, Two-sample Kolmogorov–Smirnov test, $P = 0.771$). Although there was some seasonal variation in activity, with the relative abundance of both species lowest during high water levels (Fig. 4) there was no statistical difference in the overall relative abundances grouped into 3 h bins between the four seasons for either species (Fig. 4, Kruskal-Wallis rank sum test, $P = 0.337$ and 0.438 neotropical and giant otter, respectively). Seasonal differences in the distribution of relative abundances grouped into 3 h bins were found for giant otters
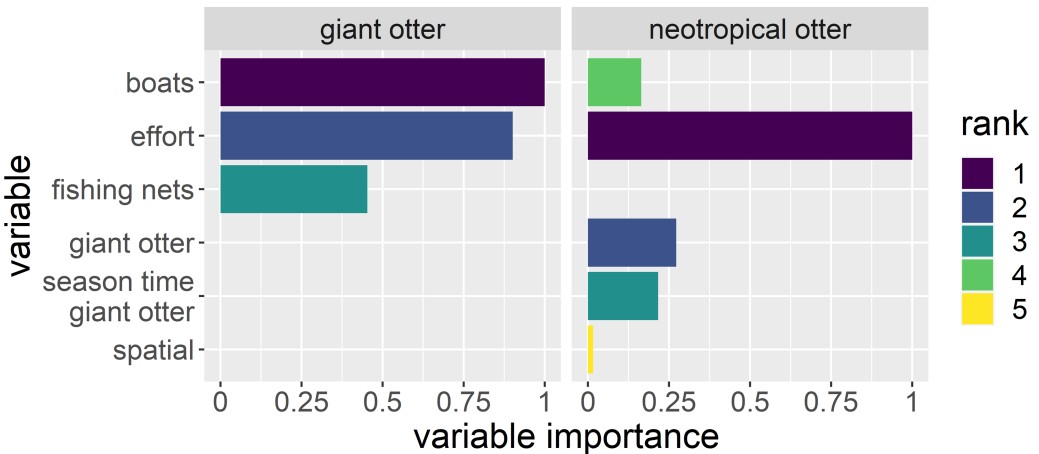

**Figure 2 Explaining patterns in giant and neotropical otter activity.** Relative importance of human river use (boats and fishing nets), season and time of day between otter species. Variables were selected from a subset of generalized additive models with Δ AIC values ≤ 6. Variable importance is the relative importance as measured by the variables' Akaike weights, with values ranging from 0 (least important) to 1 (most important). The variable "giant otter" was only included to model neotropical otter activity.

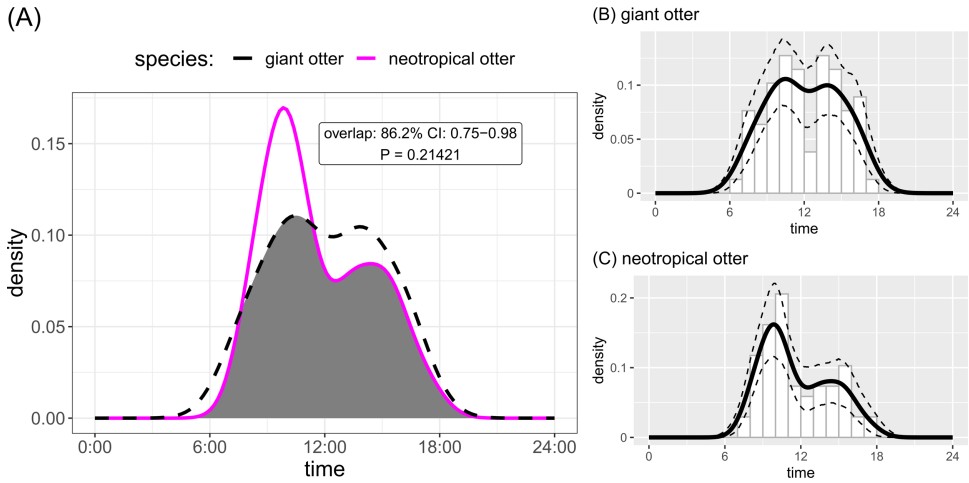

**Figure 3 Diurnal activity of sympatric otters.** Diurnal river based activity of giant and neotropical otters. (A) Overall during 53 months over nine years and (B) diurnal activity of giant otters (C) diurnal activity of neotropical otters. Sun rises at approximately 06:30 and sets at approximately 18:30 year round.

between low and high river levels (Two-sample Kolmogorov–Smirnov test, $P = 0.029$) and low and rising levels for neotropical otters (Two-sample Kolmogorov–Smirnov test, $P = 0.029$).

There appeared to be seasonal patterns to the spatial and temporal distances between species when they were seen on the same day. Both species were observed on the same day on 12 occasions. with species closer in space and time during the low river level months (Fig. 5). These same day observations occurred most frequently during low river levels,

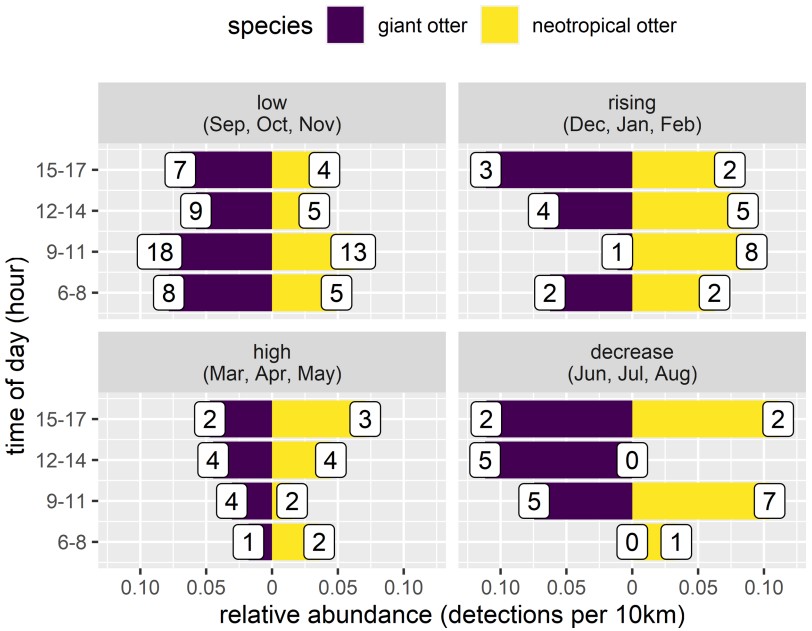

**Figure 4  Seasonal activity of sympatric otters.** Diurnal river based activity of giant and neotropical otters during different river level seasons. Value labels are sample sizes for detections during each time of day.

with both species seen on 5 days in four years (2011, 2012, 2016, 2019). Species were not usually close when seen on the same day (median distance between species 12.5 km, range 0.8 to 30.9 km), with the closest spatial difference of 808 m having a temporal separation of 8.9 h (28 April 2013, high water, 6:58 three adult giant otters and 15:49 single neotropical otter). Species were also separated temporally when seen on the same day (median time between species 3.0 h, range 14 min to 8.9 h). The temporally nearest same day registers had a difference of 14 min and 1.1 km (November 2011, low water, 9:36 single neotropical otter and 9:50 two adult giant otters).

## Diurnal activity in relation to human disturbances

The proportion of reaches with giant otters decreased threefold from 67% of the least disturbed (none/few boats no nets) to 18% of the most disturbed reaches with many boats and fishing nets. In contrast neotropical otter presence nearly doubled from 44% of the least disturbed reaches to 73% of the most disturbed reaches with fewer giant otter detections. Comparison of relative abundances also showed contrasting differences in activity between species. Giant otters were more active in reaches with less human activity, with a greater relative abundance at reaches with none or little human river use (Fig. 6, Kruskal-Wallis rank sum test $P = 0.011$). There was less variation in neotropical otter relative abundances and no significant differences among river reaches with different levels of human river use (Kruskal-Wallis rank sum test $P = 0.580$). As there were so few giant otter detections at the reaches with more boats it was not possible to compare the distribution of temporal activity among all river use classes. Giant otters tended to be more active later in the day

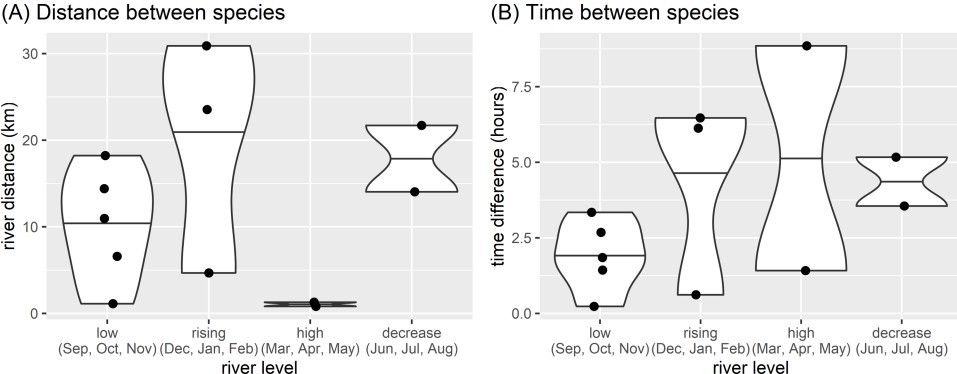

**Figure 5** **Spatial and temporal proximity.** Differences between (A) distance and (B) time of otter species detected on the same day in relation to seasonal river levels in the eastern Brazilian Amazon. Points are from 12 occasions when giant and neotropical otters were seen on the same day. Violin plots with horizontal lines showing range and median values.

at the river reaches with intermediate boat use and no nets, but there was no statistically significant difference in the distribution of activity between the few and intermediate use classes (Two-sample Kolmogorov–Smirnov test, $P > 0.940$ for all pair-wise comparisons).

There was also no statistically significant difference in relative abundance of neotropical otters in the stretches with different boat use (Fig. 6). Neotropical otters also showed no statistical difference in the timing of activity (Fig. 6, two-sample Kolmogorov–Smirnov test, $P > 0.946$ for all pairwise comparisons).

## Neotropical otter activity

There was some variation between temporal activity of neotropical otters in river reaches with and without giant otters (Fig. 7). Yet overall temporal variation was generally similar between reaches with and without giant otters (Fig. 7). Neotropical otters tended to be more active in the late afternoon (15:00–17:59) in reaches with giant otters, with relative abundances double the median value (Fig. 7A). Relative abundances also increased from 0.13 to 0.20 in reaches without and with giant otters respectively. There was no significant difference in average values or the temporal distribution of relative abundances in 3 h bins between reaches with and without giant otters (Fig. 7A, Kruskal-Wallis rank sum test, $P = 0.563$, Kolmogorov–Smirnov test, $P = 0.0596$). These similarities were reflected in the strong overlap in kernel density estimates of neotropical otter activity in reaches with and without giant otters (Figs. 7B and 7C, mean $\pm95\%$ CI [0.85, 0.73−0.98]; $P = 0.428$). There was also no significant difference in neotropical otter activity levels in reaches with and without giant otters (Figs. 7B and 7C, Wald test $P = 0.214$).

## DISCUSSION

As far as we are aware our study is the first to compare temporal and spatial patterns in the diurnal river based activity of sympatric giant and neotropical otters. Our study across a large extension of equatorial Amazonian waterways, conducted through nine years and

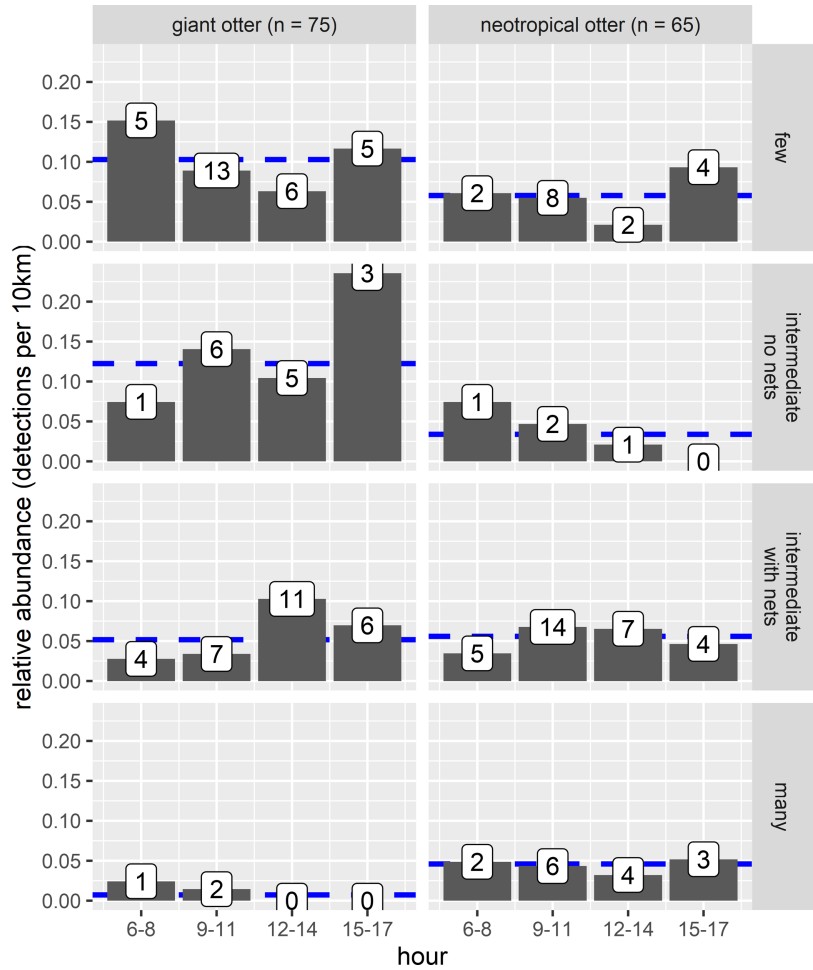

**Figure 6** **Diurnal activity of otters in relation to anthropogenic river use.** Distribution of diurnal river based activity of sympatric giant and neotropical otters in relation to human river use (boats and fishing nets). River reaches categorized by boat frequency and fishing net presence classified into four groups. Reaches with relatively few boats (less than 0.006 boats per survey km the 25% quantile of frequency values and no nets), intermediate (25–75% quantile range of frequency values) with and without nets and many boats (greater than 0.08 boats per survey km, the 75% quantile of boat frequency values all with nets). Value labels are sample sizes for detections in each time of day. Horizontal dashed blue lines show median relative abundance for each species.

based on 13,399 km of boat surveys showed (1) little difference in the timing of activity between the four river level seasons for either species; (2) that giant otters were more active in river reaches with less human activity, whereas neotropical otters showed no statistically significant difference in relative abundance in relation to human activity; and (3) in river reaches with giant otters, neotropical otters tended to be more active in the afternoon. We first discuss seasonal patterns in diurnal activity, then move to diurnal activity in relation to human disturbances and finally, we explore temporal and spatial overlap of giant and neotropical otters.

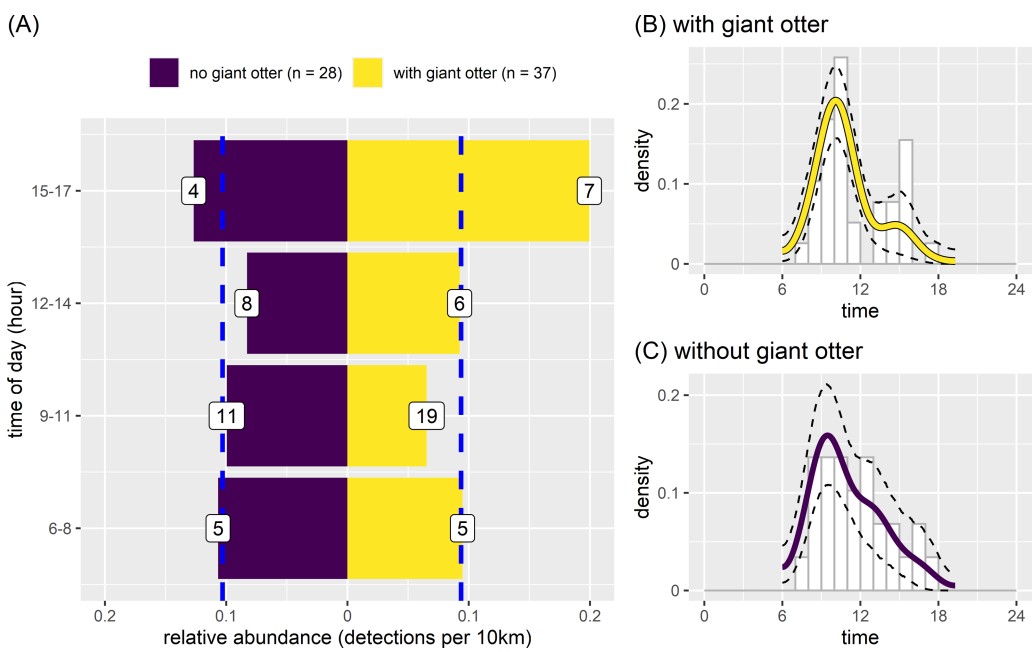

**Figure 7 Activity of neotropical otters.** Diurnal activity of neotropical otters in relation to presence of giant otters. Temporal distribution of neotropical otters in river subzones with ($n = 14$ subzones) and without ($n = 11$ subzones) the presence of giant otters. Value labels are sample sizes for detections in each time of day. Dashed blue lines show median relative abundance values in reaches with and without giant otters. Kernel density of temporal activity (B) with and (C) without giant otters.

## Seasonal patterns in diurnal activity

Our results show that diurnal activity of both otter species was not strongly affected by seasonal river level changes. While such findings require confirmation from other study sites, our results strongly suggest that any seasonal variation in river based activity is likely to be secondary to changes caused by human disturbances. There did however appear to be seasonal patterns to species coexistence, with species encountered closer in space and time during the low river level months (September-November). The small number of same day co-occurrences suggests that competition could be avoided by a combination of both spatial and temporal separation when the species occur in sympatry. Spatial and temporal separation is possible for giant and neotropical otters as both species use scent marking to maintain territories and reduce negative inter and intra-specific interactions (*Duplaix, Evangelista & Rosas, 2015*; *Rheingantz, Santiago-Plata & Trinca, 2017*). Although the small sample size limits the insight possible, increased proximity during low river levels was expected based on the relatively limited availability of aquatic feeding habitats compared with the high water level season in the study region and agrees with findings from Suriname (*Duplaix, 1980*), where both species were reported as likely using the same pools at different times to feed during the dry season.

Both species were active throughout the day across our study rivers. The general patterns in diurnal activity we found follow activity patterns reported for both species (*De Almeida & Ramos Pereira, 2017*; *Duplaix, Evangelista & Rosas, 2015*; *Rheingantz, Santiago-Plata &*

*Trinca, 2017*). The midday resting, and morning and afternoon activity peaks we found for giant otters follows very closely pattens reported from previous studies (*e.g.*, Brazilian Pantanal (*Leuchtenberger et al., 2014*) and Suriname (*Duplaix, 1980*)). The peaks of activity in the morning and reduced activity in the afternoon we found for the neotropical otter follows den use patterns reported from the Brazilian Pantanal with low human population densities (*Rheingantz et al., 2016*). Such consistency provides support for a lack of systematic bias in the results from our extensive boat surveys. River based activities in both species include hunting and movement to and from dens (*Duplaix, Evangelista & Rosas, 2015*; *Rheingantz, Santiago-Plata & Trinca, 2017*), it is therefore unsurprising that activity was observed throughout the day. Daytime river based activity has been widely documented for both species but as far as we are aware there are no similar broad scale standardized studies to enable direct comparisons with our results. Future studies including the addition of camera traps monitoring dens in our study area would be useful to facilitate comparisons with the species activity in other neotropical regions (*Leuchtenberger et al., 2014*; *Rheingantz et al., 2016*).

### Diurnal activity in relation to human disturbances

We found that giant otters were more active in river reaches with less human activity, with a greater relative abundance at reaches with none or few boats and no fishing nets. There is no evidence to suggest that differences in species activity patterns were due to differences in daylight variation as in our low latitude study area (central latitude of 0.93 N) the sun rises at approximately 06:30 am year round (day length is 11.95 h in December and 12.05 h in June). Neotropical otters showed similar patterns of diurnal river based activity across reaches with different degrees of human disturbances (boat use and with or without fishing nets). These results are not surprising as neotropical otters can be found in areas with high levels of human influence, including areas where agricultural and livestock activities occur (*Pardini & Trajano, 1999*; *Rheingantz, Santiago-Plata & Trinca, 2017*; *Rheingantz et al., 2011*; *Smith et al., 2020*).

As nocturnal activity has been recorded in both species (*Duplaix, Evangelista & Rosas, 2015*; *Rheingantz, Santiago-Plata & Trinca, 2017*) additional data from complementary methods such as camera traps and/or telemetry would generate additional insight across the 24 h period. Activity (diurnal and nocturnal) generally occurs close to the dens of both species (*Evangelista & Rosas, 2011*; *Leuchtenberger et al., 2014*; *Rheingantz et al., 2016*), and a previous study found no giant otter dens in the areas most intensely used by humans in the study area (*Oliveira, Norris & Michalski, 2015*). It is therefore unlikely that the absence of diurnal observations was due to a temporal change in giant otter activity in the most disturbed river reaches. The most plausible explanations are that giant otters have either moved away from and/or been hunted out of areas more intensely used by humans (*Raffo et al., 2022*). While our data provides a broad scale assessment, it is likely that giant otters with complex social interactions will also demonstrate population and individual level changes to human disturbances. For example, studies with ecotourism suggest giant otters can adapt to well regulated low intensity human disturbances (*Barocas et al., 2022*). It is possible that changes in activity and other behaviors to avoid humans could be facilitated

by social learning in giant otters (*Barocas et al., 2022*; *Davenport, 2010*; *Schmelz et al., 2017*). But, to date there have been very few long term behavioral studies of wild groups that are needed to confirm such a theory.

## Neotropical otter activity

Our findings support those from previous studies that suggest no direct competition between giant and neotropical otters. As far as we are aware there have been no reports of direct interspecific competition between any otter species. Indeed, sympatry facilitated by habitat, trophic and temporal separation has been widely documented between otter species across the globe, with sympatry in up to three otter species found in Thailand (*Ebensperger & Botto-Mahan, 1997*; *Krupa, Borker & Gopal, 2017*; *Kruuk et al., 1994*; *Lélias, Lemasson & Lodé, 2021*; *Somers & Purves, 1996*). We found giant and neotropical otters were on average approximately 12 km and 3 h apart when detected on the same day, which indicates that competition is avoided both temporally and spatially. As suggested by previous studies our findings support the idea that neotropical otters can be more active later in the day in more disturbed areas (*Rheingantz et al., 2016*), but that human disturbances do not appear to generate changes at the same scale as the sympatric giant otter.

The reduced presence and stable relative abundances in reaches with fewer boats and nets suggests that there are other factors limiting diurnal river use by neotropical otters in these relatively undisturbed reaches. There are several non-mutually exclusive explanations for why neotropical otters did not show increased activity as expected in least disturbed reaches. We suggest the most plausible was the presence of giant otters. Unlike the giant otter, neotropical otters did not increase activity in reaches with few human disturbances. It was expected that neotropical otters would increase activity (*i.e.,* "if" and "when" they were active) where there was less human disturbances (*Rheingantz, Santiago-Plata & Trinca, 2017*). The neotropical otter is more generalist with a much broader diet (including diverse small bodied vertebrate species) and less restrictive habitat requirements as they are found across diverse tropical and terrestrial aquatic habitats from Mexico to Argentina (*De Almeida & Ramos Pereira, 2017*; *Rheingantz, Santiago-Plata & Trinca, 2017*). As such this more generalist species is not expected to show such strong broad scale responses to environmental changes across 218 km of Amazonian rivers as the more sensitive giant otter. As giant otters were recorded in the reaches with fewer human disturbances, we can therefore safely assume that these reaches also had suitable habitat and prey availability for the more generalist neotropical otters. The most plausible explanation for the observed broad-scale differences in neotropical activity is that neotropical otters may use the main river channels less in the presence of giant otters and use more the smaller perennial inland streams (typically less than 2 m wide and 1 meter deep) that were not possible to survey with motorized boats.

In the case of our study, neotropical otter activity increased between 15–17 in river reaches used by giant otters. These findings support those from perhaps the earliest study showing sympatry that suggested neotropical otters could be more crepuscular and more nocturnal when sympatric with giant otters (*Duplaix, 1980*). Nocturnal activity by neotropical otters has been recorded along the study rivers (*Michalski et al., 2021*) and

future studies could focus on comparing den activity recorded by camera traps in river reaches with and without giant otters to test if nocturnal activity of neotropical otters also changes.

Although human disturbances may increase temporal overlap among terrestrial carnivores (*Sévêque et al., 2020*; *Sévêque et al., 2021*), our findings show that human river use dramatically reduced giant otter activity and therefore reduced spatial overlap with neotropical otters. It is possible that the absence of the larger bodied giant otter could enable neotropical otters to remain active for longer. This could lead to increased feeding opportunities, which could result in population increases and expansion. At the same time any increased neotropical otter activity could also increase negative interactions along rivers used by humans. We are not aware of studies quantifying such dynamics in neotropical otters. Establishing how anthropogenic impacts that reduce the occurrence of giant otters may subsequently influence activity, trophic dynamics and populations of sympatric neotropical otters should be a priority for future studies.

A potential caveat to our findings is that our analysis specifically focuses on broad scale river based activity patterns. Both species also use terrestrial habitats to different degrees and the dynamics of river and terrestrial activity in response to disturbances remains unexplored. Additionally, the use of boat surveys means that it was more likely to detect the larger and less secretive giant otter. Neotropical otters are smaller and solitary, which makes them harder to detect from boats. Here we assume that such differences in detectability do not bias our broad scale comparisons among river reaches. However, more detailed studies are required to determine more localized responses at the scale of individuals and/or extended family groups in the case of giant otters.

## CONCLUSIONS

Our findings support evidence that activity in both otter species is flexible, with daytime activity changing due to human disturbances in the case of giant otters and the presence of the larger bodied sympatric species in the case of neotropical otters. Giant otters reduced activity in river reaches used by humans and this suggests that intensification and expansion of river use by humans may have a far greater impact on giant otters across their range than previously thought.

## ACKNOWLEDGEMENTS

The Instituto Chico Mendes de Conservação da Biodiversidade (ICMBio) and the Federal University of Amapá (UNIFAP) provided logistical support. We are grateful to Alvino Pantoja Leal, Cremilson and Cledinaldo Alves Marques, Edinaldo and Davi Sousa, and Gilberto Souza for their invaluable assistance during fieldwork. We also thank all students, volunteer interns, and field assistants who participated on field activities over the study period.

### Funding

This research was funded by the National Academy of Sciences and the United States Agency for International Development through the Partnership for Enhanced Engagement in Research (award number AID-OAA-A11-00012) to Darren Norris and Fernanda Michalski. This research was also supported by Conservation International–Brazil and the Walmart Institute–Brazil through the project "Support to the implementation of the Amapá National Forest", the Rufford Small Grants for Nature Conservation, the CNPq (Processes 477629/2011-3, 301562/2015-6, 403679/2016-8, 302806/2018-0), and the Conservation, Food & Health Foundation to Fernanda Michalski. The funders had no role in study design, data collection and analysis, decision to publish, or preparation of the manuscript.

### Grant Disclosures

The following grant information was disclosed by the authors:
National Academy of Sciences and the United States Agency for International Development through the Partnership for Enhanced Engagement in Research: AID-OAA-A11-00012.
Conservation International–Brazil and the Walmart Institute–Brazil through the project "Support to the implementation of the Amapá National Forest", The Rufford Small Grants for Nature Conservation, the CNPq: Processes 477629/2011-3, 301562/2015-6, 403679/2016-8, 302806/2018-0.
Conservation, Food & Health Foundation to Fernanda Michalski.

### Competing Interests

Fernanda Michalski is associated with Pro-Carnivores Institute. Fernanda Michalski is an Academic Editor for PeerJ. Darren Norris is an Academic Editor for PeerJ. The authors declare there are no competing interests.

### Author Contributions

- Darren Norris conceived and designed the experiments, performed the experiments, analyzed the data, prepared figures and/or tables, authored or reviewed drafts of the article, and approved the final draft.
- Fernanda Michalski conceived and designed the experiments, performed the experiments, prepared figures and/or tables, authored or reviewed drafts of the article, and approved the final draft.

### Animal Ethics

The following information was supplied relating to ethical approvals (*i.e.*, approving body and any reference numbers):

This study used data from non-invasive field observations and did not involve direct contact or interactions with animals.

## Field Study Permissions

The following information was supplied relating to field study approvals (*i.e.*, approving body and any reference numbers):

Fieldwork was conducted under research permit numbers IBAMA/SISBIO 26653, 49632, and 69342 to DN and FM, issued by the Instituto Chico Mendes de Conservação da Biodiversidade (ICMBio).

## Data Availability

The raw data are available in the Supplementary File.

## Supplemental Information

Supplemental information for this article can be found online at http://dx.doi.org/10.7717/peerj.15742#supplemental-information.

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
