# Peer review of "Human disturbances and the daytime activity of sympatric otters along equatorial Amazonian rivers"

_PeerJ, doi:10.7717/peerj.15742_

## Round 0.1 · original submission · Major Revisions

First of all, my apologies for the time it took to send a decision about your article. Most potential reviewers were engaged with fieldwork.

Both reviewers agree that your article can be published after major revision. They provided an excellent and comprehensive set of suggestions that you should use to improve the manuscript. Give special attention to the statistical treatment of your data and how you present your arguments.

Please ensure that all review and editorial comments are addressed in a response letter.

I look forward to seeing an improved version.

Reviewer 1 ·

Basic reporting

The basic reporting of this article is generally clear, although there are few places where sentence structures can be improved for clarity (see detailed comments on the annotated pdf). The overall article structure is sufficient, raw data has been shared and the paper is self-contained.

Experimental design

The overall experimental design is relatively straight forward and took the form of boat-based surveys for the two species of otter that occur sympatrically in the study region. The authors collected a substantial amount of data during the course of the study, which is certainly worth publishing. I thought the methods were described in sufficient detail and the overall research questions were clear.

Validity of the findings

Here I have some concerns about the statistical treatment of the data and the conclusions reached. Firstly, the authors make many inferences from evident patterns in the data that is not supported by statistically significant results.

But perhaps more importantly, my feeling is that the overall treatment of the dataset can be improved and the current repeated hypothesis testing on a single dataset is also not ideal, particularly without the application of any appropriate corrections. The analyses ‘as is’ are prone to increased chances of Type 1 errors, and furthermore probably suffers from spatial autocorrelation issues.

I think the collected dataset is better suited to a binomial generalised linear model approach (the authors may have to look at options to account for zero-inflation in this context), or similar, whereby the authors can potentially generate a single model that assesses the relative influence of the various predictors (e.g. boat density, nets and season) on the probability of encountering either of the two species. I would encourage the authors to re-evaluate the statistical approach altogether with this in mind.

Additional comments

Please see a few more specific comments as annotations on the manuscript (attached).

Annotated reviews are not available for download in order to protect the identity of reviewers who chose to remain anonymous.

Reviewer 2 ·

Basic reporting

I read the manuscript titled “Human disturbances and the daytime activity of sympatric otters along equatorial Amazonian rivers” carefully. The authors describe an extensive and impressive survey effort carried out in the Brazilian Amazon to detect two species of otters and understand whether there are interactions between these species, and whether their distribution is influenced by human activity. The manuscript deals with two carnivore species that are significant parts of these riverine ecosystems and examines important ecological questions and has some positive aspects.
One major issue is that some of the data in the manuscript, namely on giant otters, has been published in the past by some of the same authors (de Oliveira et al. 2015). They should be more transparent on what is the novelty of this work by stating this specifically. I have a number of additional issues with the manuscript as it is currently written.
1. Some background is missing, especially regarding the influence of anthropogenic activity on the two otter species.
2. The motivation for the research is not well articulated and the hypotheses and predictions are not clearly stated. Combined with the fact that part of the data was previously published, it is unclear whether and why we need more of the same findings.
3. The statistical methodology used and descriptive graphic presentation of the results do not fit the data structure.
4. I feel that, while the authors generally use cautious language, some of the inference does not follow form the body of data found, especially regarding the interactions and co-occurrence between the otter species.
I feel that these aspects could be improved upon, and the manuscript could be revised.

More specific comments appear below.
Introduction
Lines 60-67 – The authors only mention the two species by name and provide no background data on their ecology or even difference in size and diet. The authors should elaborate on the relevant findings on the distribution and activity of the two otter species, and why they suspect that these species would compete or have any ecological interactions. Also, because one of the author’s main questions is whether these species are influenced by human activities, they should provide a more complete overview of the current evidence.
Line 95-98 – The research questions are sloppily articulated and some of them do not follow from the background provided. I feel that the motivation to the study is an important part of any scientific work, and the authors should invest more effort on explaining the rationale behind their research questions.
Methods
Lines 135-147 – The survey procedures are well described but crucial details regarding the study design are missing: How may times was each river segment surveyed? Was there an effort to control for time of day? (i.e. if the boat departs from the same location in the morning each year, would this not introduce bias in the detection probability?).
General comment – The authors discuss how they divided the data but do not describe the statistical procedures. From the results, it appears that they used two-sampled/non-parametric tests that are not suited for the data structure. In my opinion, given that some observations are repeated and thus there is dependence in the data, linear mixed models (Harrison et al. 2018) are a more conceptually sound choice in this case. The authors have demonstrated their ability to use such models in the past with a similar dataset (de Oliveira et al. 2015).
Results
Line 209 – 220 – I think the small number of days (12) when the two species were observed, the spatial distance and the time windows found do not allow inference regarding their interactions. I have no problem with how the authors describe it here but feel that very limited inference can be drawn from these data.
Line 239 – 241 – It cannot be understood from Figures 4 and 5 that “neotropical otters were less active in river reaches with giant otters”. In addition, Figure 6 suggests an opposite pattern. From the data presented, it does not follow that giant otters suppress neotropical otters.
Discussion
Line 303 – This data resolution does not demonstrate active avoidance of humans (which in my interpretation is movement away from human activity when it is evident in the landscape). It is more consistent with selection of habitats/ home ranges that are undisturbed.
Line 310 – 320 – I feel that the small number of detections and the limited resolution of daily co-occurrences between the two otter species make most of the interpretation here highly speculative. Assertions like “human disturbances do not appear to generate changes at the same scale as the sympatric giant otter” are not supported by the results.
Line 345-350 – See my previous comments regarding what can and cannot be inferred from the results.
References
Harrison, X. A., L. Donaldson, M. E. Correa-Cano, J. Evans, D. N. Fisher, C. E. D. Goodwin, B. S. Robinson, D. J. Hodgson, and R. Inger. 2018. A brief introduction to mixed effects modelling and multi-model inference in ecology. PeerJ 2018:1–32.
de Oliveira, I. A. P., D. Norris, and F. Michalski. 2015. Anthropogenic and seasonal determinants of giant otter sightings along waterways in the northern Brazilian Amazon. Mammalian Biology 80:39–46.

Experimental design

.

Validity of the findings

.

Additional comments

None

---

## Round 0.2 · accepted · Accept

Overall the manuscript is written well and communicates a very good study. The comments from both reviewers were positive. The only minor changes requested were from a misunderstanding of the interaction term giant otter and season time. Congratulations and have a great day!

Reviewer 1 ·

Basic reporting

Just some minor suggestions - please see attached annotated manuscript.

Experimental design

No comment

Validity of the findings

No comment

Additional comments

Thank you to the authors for their careful consideration of my comments and their thorough feedback. I only have a few minor suggestions and corrections that I have indicated as annotations on the attached version.

Annotated reviews are not available for download in order to protect the identity of reviewers who chose to remain anonymous.

Reviewer 2 ·

Basic reporting

I read the revised manuscript titled “Human disturbances and the daytime activity of sympatric otters along equatorial Amazonian rivers” carefully.
The authors have done a remarkable job addressing the issues previously raised by reviewers and added relevant analyses and graphics. As a result, the manuscript is much stronger compared to the previous version.

As a result, the manuscript is much improved and I have no further objections to its acceptance.

Experimental design

1. The background on the impacts of human activity on relevant species is more extensive and the hypotheses are better developed.
2. The authors have improved the statistical methodology. They added a GAM multivariate analysis to help examine relative support of drivers of activity for both otter species. They also used a kernel estimator to better model and represent activity patterns for both species.

Validity of the findings

3. The authors have added important information the Discussion, making it more organized and providing better justification for their insights. The revised version of the Discussion also elaborates more on the limitations of data collected by the authors.

Additional comments

No comment